# IL-10 and class 1 histone deacetylases act synergistically and independently on the secretion of proinflammatory mediators in alveolar macrophages

**Brent A. Stanfield**[1,2,3], **Todd Purves**[1,2,4], **Scott Palmer**[1,5,6], **Bruce Sullenger**[1,2], **Karen Welty-Wolf**[1,5], **Krista Haines**[1,2,3], **Suresh Agarwal**[1,2,3], **George Kasotakis**[1,2,3]*

**1** Duke University Medical Center, Durham, North Carolina, United States of America, **2** Department of Surgery, Duke University Medical Center, Durham, North Carolina, United States of America, **3** Division of Trauma, Acute and Critical Care Surgery, Duke University Medical Center, Durham, North Carolina, United States of America, **4** Division of Urology, Duke University Medical Center, Durham, North Carolina, United States of America, **5** Department of Medicine, Duke University Medical Center, Durham, North Carolina, United States of America, **6** Duke Clinical Research Institute, Durham, North Carolina, United States of America

* george.kasotakis@duke.edu

**Data Availability Statement:** All relevant data are within the paper.

## Abstract

### Introduction

Anti-inflammatory cytokine IL-10 suppresses pro-inflammatory IL-12b expression after Lipo-polysaccharide (LPS) stimulation in colonic macrophages, as part of the innate immunity Toll-Like Receptor (TLR)-NF-κB activation system. This homeostatic mechanism limits excess inflammation in the intestinal mucosa, as it constantly interacts with the gut flora. This effect is reversed with Histone Deacetylase 3 (HDAC3), a class I HDAC, siRNA, suggesting it is mediated through HDAC3. Given alveolar macrophages' prominent role in Acute Lung Injury (ALI), we aim to determine whether a similar regulatory mechanism exists in the typically sterile pulmonary microenvironment.

### Methods

Levels of mRNA and protein for IL-10, and IL-12b were determined by qPCR and ELISA/Western Blot respectively in naïve and LPS-stimulated alveolar macrophages. Expression of the NF-κB intermediaries was also similarly assessed. Experiments were repeated with AS101 (an IL-10 protein synthesis inhibitor), MS-275 (a selective class 1 HDAC inhibitor), or both.

### Results

LPS stimulation upregulated all proinflammatory mediators assayed in this study. In the presence of LPS, inhibition of IL-10 and/or class 1 HDACs resulted in both synergistic and independent effects on these signaling molecules. Quantitative reverse-transcriptase PCR on key components of the TLR4 signaling cascade demonstrated significant diversity in IL-

**Funding:** The author(s) received no specific funding for this work.

**Competing interests:** The authors have declared that no competing interests exist.

10 and related gene expression in the presence of LPS. Inhibition of IL-10 secretion and/or class 1 HDACs in the presence of LPS independently affected the transcription of MyD88, IRAK1, Rela and the NF-κB p50 subunit. Interestingly, by quantitative ELISA inhibition of IL-10 secretion and/or class 1 HDACs in the presence of LPS independently affected the secretion of not only IL-10, IL-12b, and TNFα, but also proinflammatory mediators CXCL2, IL-6, and MIF. These results suggest that IL-10 and class 1 HDAC activity regulate both independent and synergistic mechanisms of proinflammatory cytokine/chemokine signaling.

## Conclusions

Alveolar macrophages after inflammatory stimulation upregulate both IL-10 and IL-12b production, in a highly class 1 HDAC-dependent manner. Class 1 HDACs appear to help maintain the balance between the pro- and anti-inflammatory IL-12b and IL-10 respectively. Class 1 HDACs may be considered as targets for the macrophage-initiated pulmonary inflammation in ALI in a preclinical setting.

## Introduction

Acute Respiratory Distress Syndrome (ARDS) is a clinical syndrome characterized by an exaggerated immune response in the lungs to local or systemic inflammatory stimuli, and manifests with severe hypoxia necessitating prolonged ventilatory support and critical care [1]. Acute Lung Injury (ALI) describes collectively the pathologic changes observed in ARDS lungs. Over 10% of all ICU patients meet criteria for ARDS [2] and more than 200,000 cases are diagnosed annually in the U.S. [1], costing over $434,000 per hospital stay [3]. Despite advances in critical care, no effective treatments for ARDS exist at this time, and non-specifically targeted anti-inflammatory therapies have largely failed to improve mortality [4–9], which approximates 40% [2]. Novel, highly targeted therapies for ARDS are thus sorely needed, yet crucial factors limiting their development include our poor understanding of the underlying pathophysiology, and the heterogeneous etiology of the syndrome.

Bacterial pneumonia is by far the most common cause of ARDS [2], and the alveolar macrophage (AM), one of the chief Antigen Presenting Cells (APC) in the lung parenchyma, plays a key role in the initiation and perpetuation of the maladaptive innate immune response that leads to ALI, after interaction with pathogenic components [10]. Specifically in pneumonia, Pathogen-Associated Molecular Patterns (PAMPs) bind to TLR, which in turn activate the NF-κB axis. This leads to transcription of both proinflammatory, such as IL-12, and anti-inflammatory cytokines, such as IL-10, the balance between whom regulates the ebb and flow phases of the acute innate immune response [11].

It has been shown that both basally, and activated with exposure to enteric bacteria, IL-10 is regulated via the MyD88 pathway in wild-type, pathogen-free intestinal macrophages, and this, in turn, restricts IL-12b synthesis. Class 1 HDACs include HDAC1, HDAC2, and HDAC3. Specifically, HDAC3 has been described as a key mediator of IL-12b transcription. Mechanistically, IL-10 inhibits IL-12b synthesis by modulating HDAC3 activity at the IL-12b promoter. HDAC3 represses IL-12b transcription by deacetylation of key histones within its promoter. This shifting of the balance by IL-10 to an anti-inflammatory state in colonic macrophages is responsible for the immune tolerance of these APCs to the abundant intestinal

flora [12]. However, little is known about the contribution of IL-10 and the AM to the immune homeostasis in the typically pathogen-free pulmonary parenchyma.

With this project we aim to determine the MyD88- dependent or independent pathway that regulates the IL-10 –IL-12b balance, and the role of class 1 HDACs in the regulation of IL-12b synthesis in alveolar macrophages.

## Materials and methods

### Drugs

Agent MS275, a selective class 1 HDAC inhibitor [13] (Entinostat, Santa Cruz Biotechnology, sc-279455), was reconstituted to a stock concentration of 10 mM in DMSO. The immunomodulator AS101 (Santa Cruz Biotechnology, sc-203825) that inhibits IL-10 protein synthesis [14] was reconstituted at a concentration of 5 mM in DMSO.

### Cell lines and culture conditions

Murine alveolar macrophage cells (MH-S) were purchased from the American Type Tissue Culture Collection (ATCC, CRL-2019). Cells were maintained in RPMI-1640 media (Thermo-Fisher, 11875135) containing 10% heat inactivated fetal bovine serum (HyClone, SH30071.03HI L-Glutamine-Penicillin-Streptomycin solution 1:100 (Sigma, G6784), and 1x 2-mercaptoethanol (Gibco, 31350010). Cells were cultured in 5% $CO_2$ at 37˚C. Twenty-four hours prior to stimulation MH-S cells were trypsinized using 0.25% trypsin-EDTA and plated at a density of 1E6 cells/well in a 6-well plate. Cells were then stimulated with LPS at a final concentration of 100 ng/mL and simultaneously treated with either AS101 300 nM, MS275 10 μM, or a combination of AS101 (300 nM) and MS275 (10 μM). Cells were then cultured under these conditions for an additional 24 hours and sampled for analysis.

### Sampling

Total RNA was collected using the RNeasy mini-kit (Qiagen, 74104) as per the manufacturer's instructions and eluted in a final volume of 60 uL RNase and DNase free water. RNA samples were frozen at -80˚C until analysis. Total cell lysates were prepared by washing cells 3 times in 1x PBS and lysed in complete lysis buffer (NaCl, 150 mM, Triton-X100 1.0%, Tris-Cl (50 mM, pH 8.0) containing proteinase (Roche, 11836170001) and phosphatase (Roche, 04906845001) inhibitor cocktails. Whole cell lysates were frozen at -80˚C until analysis. Tissue culture supernatants were collected at 24 hours post stimulation and frozen in 1 mL aliquots at -80˚C until analysis.

### cDNA synthesis and quantitative PCR

Copy DNA was synthesized from total RNA using the Applied Biosystems High-Capacity cDNA Reverse Transcription Kit (ThermoFisher, 4368814) as per the manufacturer's instructions. cDNA was then diluted 1:10 in nuclease free water and used for qPCR analysis using PowerUP SYBR Green Master Mix using the manufacturers cycling temperatures with the indicated primers (Table 1) on a Bio-Rad CFX96 Touch Real-Time PCR Detection System.

### Western blot

Thirty micrograms of whole cell lysates were reduced in a mixture of 1x LDS sample Buffer (TruPAGE, PCG3009) and 50 μM DTT at 80˚C for 10 minutes. Reduced lysates were then ran on a 4–15% Criterion TGX Stain-Free Protein Gel (BioRad, 5678083) in 1x tris/glysine/SDS (BioRad, 1610732) at 150 v for 1 hour and transferred to a nitrocellulose membrane in 1x tris/

**Table 1. Primers used in this study.**

| Target | PrimerBank ID | Sequence | Length | Tm | Amplicon |
|---|---|---|---|---|---|
| Actb | 6671509a1 | GGCTGTATTCCCCTCCATCG | 20 | 61.8 | 154 |
| | | CCAGTTGGTAACAATGCCATGT | 22 | 61.1 | |
| HSP90 | 28277018a1 | GTCCGCCGTGTGTTCATCAT | 20 | 62.8 | 168 |
| | | GCACTTCTTGACGATGTTCTTGC | 23 | 62.4 | |
| IkK | 6680942a1 | GTCAGGACCGTGTTCTCAAGG | 21 | 62.3 | 118 |
| | | GCTTCTTTGATGTTACTGAGGGC | 23 | 61.4 | |
| IL-10 | 6754318a1 | GCTCTTACTGACTGGCATGAG | 21 | 60.2 | 105 |
| | | CGCAGCTCTAGGAGCATGTG | 20 | 62.7 | |
| Il-12b | 6680397a1 | TGGTTTGCCATCGTTTTGCTG | 21 | 62.3 | 123 |
| | | ACAGGTGAGGTTCACTGTTTCT | 22 | 61.2 | |
| IRAK1 | 13435858a1 | CCAGAGGCAAAACTCCCAACA | 21 | 62.5 | 61 |
| | | AGAGCACCTCCCCAAATAGAG | 21 | 60.7 | |
| IRAK4 | 23943898a1 | CATACGCAACCTTAATGTGGGG | 22 | 61.3 | 125 |
| | | GGAACTGATTGTATCTGTCGTCG | 23 | 60.7 | |
| MYD88 | 26354939a1 | TCATGTTCTCCATACCCTTGGT | 22 | 60.5 | 175 |
| | | AAACTGCGAGTGGGGTCAG | 19 | 61.9 | |
| p50 | 30047197a1 | ATGGCAGACGATGATCCCTAC | 21 | 61.1 | 111 |
| | | TGTTGACAGTGGTATTTCTGGTG | 23 | 60.4 | |
| RELA | 6677709a1 | AGGCTTCTGGGCCTTATGTG | 20 | 61.6 | 111 |
| | | TGCTTCTCTCGCCAGGAATAC | 21 | 61.6 | |
| RIP1 | 34328467a1 | GAAGACAGACCTAGACAGCGG | 21 | 61.6 | 182 |
| | | CCAGTAGCTTCACCACTCGAC | 21 | 62.1 | |
| TAK1 | 27881429a1 | CGGATGAGCCGTTACAGTATC | 21 | 60 | 168 |
| | | ACTCCAAGCGTTTAATAGTGTCG | 23 | 60.6 | |
| TRAF6 | 6678429a1 | AAAGCGAGAGATTCTTTCCCTG | 22 | 60 | 125 |
| | | ACTGGGGACAATTCACTAGAGC | 22 | 61.4 | |
| TRIF | 23272109a1 | AACCTCCACATCCCCTGTTTT | 21 | 61.3 | 81 |
| | | GCCCTGGCATGGATAACCA | 19 | 61.8 | |

glycine (BioRad, 1610734) containing 20% methanol for 1 hour. The membrane was then washed in 1x TBS-T and blocked for 1 hour in 5% BSA-TBS-T. Primary mouse anti-IκBα (Novus, NB100-56507), mouse anti-phospho-IκBα S32/36 (Cell Signaling Technology, 9246S), and mouse anti-GAPDH (Biolegend, 607902) was then diluted to 1 µg/mL in 5% BSA-TBS-T and incubated at 4˚C overnight with agitation. The membrane was then washed 3x for 10 minutes per wash in 1x TBS-T and secondary goat anti-mouse HRP (Abcam, ab205719) diluted 1:1000 in 5% BSA-TBS-T was incubated at room temperature for 1 hour after which the membrane was washed 3x for 10 minutes per wash in 1x TBS-T and visualized using SuperSignal West Femto Maximum Sensitivity Substrate (Thermo Scientific, 34094) on a BioRad Chemi-Doc MP Imaging System.

## Fluorescent microscopy and image processing

22 x 22 mm glass microscope cover slips (VWR, 16004–302) were autoclaved and then coated with 0.1 mg/mL poly-d lysine hydrobromide in $H_2O$ (Sigma, P7280-5MG) overnight at 4˚C with agitation in 6 well plates (Sigma, CLS6516). Following coating, cover slips were then washed 3 times in sterile 1x PBS (Gibco, 10010–023). MH-S cells were cultured directly on the cover slips by seeding at 1E5 cells/well and incubating overnight at 5% $CO_2$ at 37˚C and

stimulated with LPS at a final concentration of 100 ng/mL and simultaneously treated with either AS101 300 nM, MS275 10 μM, or a combination of AS101 (300 nM) and MS275 (10 μM). Cells were then cultured under these conditions for an additional 24 hours, washed 3 times in 1x PBS, and fixed for 10 minutes at room temperature in 4% paraformaldehyde (Electron Microscopy Sciences, 15710-S) in PBS. Following fixation, samples were washed 3 times in 1x PBS and permeabilized by incubating samples in 0.1% Triton X-100 (Sigma, T8787) in PBS for 10 minutes at room temperature and washed 3 times for 5 minutes each at room temperature in 1x PBS. Samples were then blocked for 30 minutes at room temperature with blocking solution (1% bovine serum albumin and 22.52 mg/mL glycine in PBS + 0.1% Tween). Samples were then stained with mouse anti-RelA (Novus, NB100-56172), and with isotype controls mouse IgG1 (R&D Systems, MAB002), diluted 1:100 in 1% BSA PBS-T overnight at 4°C. Samples were then washed 3 times for 5 minutes each at room temperature in 1x PBS-T and secondary goat anti-mouse IgG H&L Alexa Fluor 488 (Abcam, ab150113) was diluted 1:1000 in 1% BSA PBS-T and incubated at room temperature for 1 hour. Samples were then washed 3 times for 5 minutes each at room temperature in 1x PBS-T and mounted on microscope slides with Vectashield antifade mounting media with DAPI (Vector Laboratories, H-1200). Images were taken on a Ziess Axio Imager Widefield Fluorescence Microscope using the 40x objective and an Axiocam 506 monochromatic camera. Three random fields were imaged and subjected to post image processing in imageJ/Fiji [15, 16]. Nuclear levels of RelA were determined by using the ImageJ Intensity Ratio Nuclei Cytoplasm Tool [17, 18], where DAPI was used as the nuclear stain. The threshold, select area, and ROI manager functions of ImageJ were used to reduce background as described previously [19].

## ELISA

One hundred microliters of cell culture supernatant were used in each quantitative ELISA looking at the amount of IL-10 (Biolegend, 431414), IL-12b (Biolegend, 431604), TNFa (Biolegend, 430904), CXCL2 (R&D Systems, DY452-05), IL-6 (Biolegend, 431304), and MIF (Biolegend, 444107) as per the manufacturer's instructions an read on a Tecan infinite M200 Pro plate reader at the required absorbance.

## Results

### IL-10 and class 1 HDACs independently and synergistically regulate IL-12b transcription in alveolar macrophages

LPS stimulation has been extensively shown to rapidly induce TLR4 activation in macrophages [20–23] and previous work has described the downstream regulatory role IL-10 plays on the expression of IL-12b through the activation of HDAC3 in colonic macrophages [12]. Here, we sought to define a tissue specific phenotype of the IL-10—IL-12 axis and expand our knowledge on the immunomodulatory properties of altering IL-10 synthesis and inhibition of class 1 HDACs under inflammatory conditions in alveolar macrophages. Of note, IL-10 was constitutively expressed in naïve states, while IL-12b requires inflammatory stimulation. Here we demonstrate that in the presence of LPS, AS101 inhibited IL-10 protein synthesis, but not transcription (**Figs 1A and 5A**). LPS stimulation increased the transcription of IL-12b in alveolar macrophages when compared to unstimulated controls. Inhibition of both IL-10 synthesis and class 1 HDACs independently increased IL-12b transcription over LPS alone and together acted synergistically increasing IL-12b transcription ~35k fold over unstimulated controls (**Fig 1B**). Interestingly, inhibition of IL-10 synthesis by AS101 [14] enhanced IL-10 transcription, while class 1 HDAC inhibition by MS-275 had the same effect. The combination of both

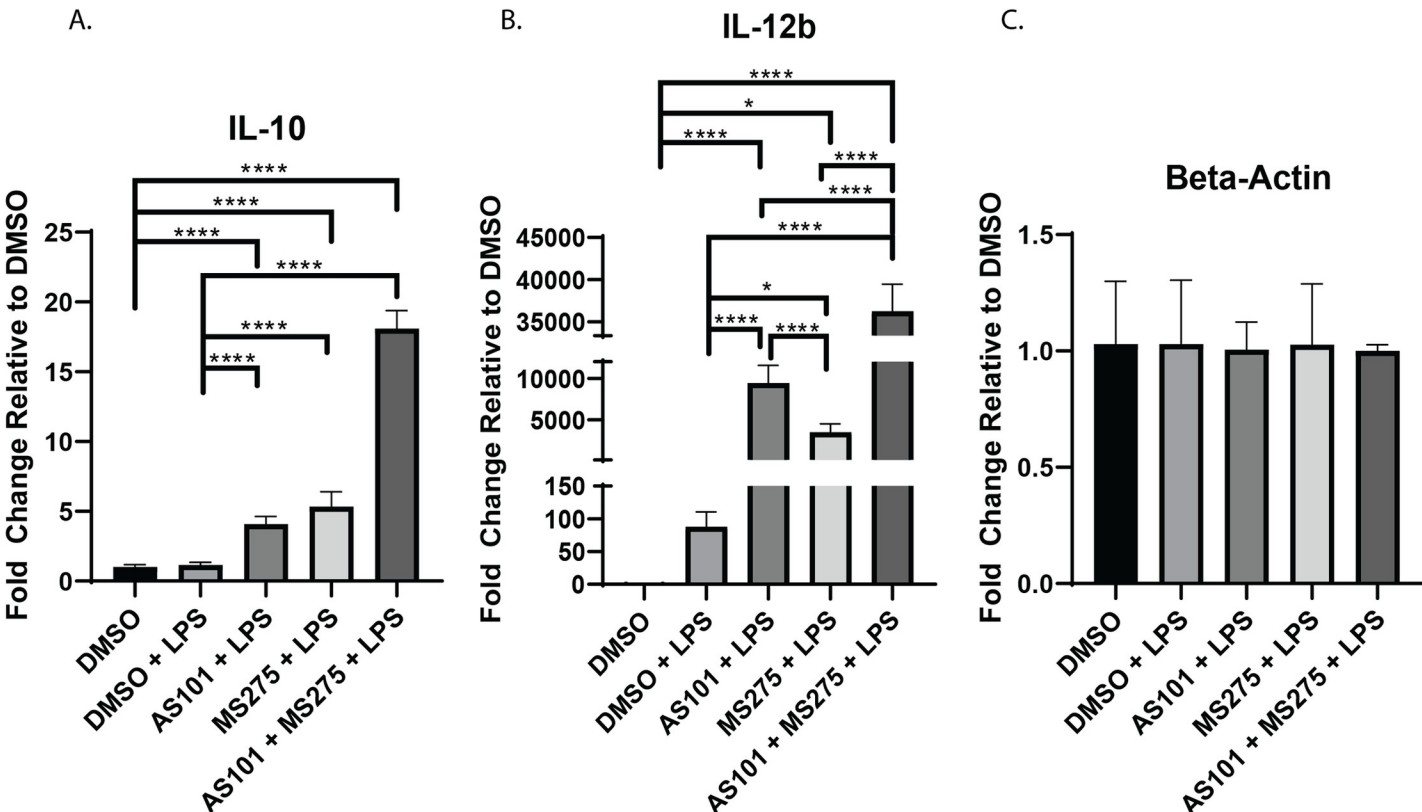

**Fig 1. Inhibition of either IL-10 synthesis or class 1 HDACs increases IL-12b transcription.** Quantitative rtPCR analysis of RNA extracted from murine alveolar macrophage cells (MH-S) treated with DMSO, treated with DMSO and stimulated with 100 ng/mL LPS (DMSO + LPS), treated with 300 nM AS101 and stimulated with 100 ng/mL LPS (AS101 + LPS), treated with 10 μM MS275 and stimulated with 100 ng/mL LPS (MS275 + LPS), and treated with both 300 nM AS101 and 10 μM MS275 and treated with 100 ng/mL LPS (AS101 + MS275 + LPS). Relative quantities of A. IL-10, B. IL-12b, and C. Beta-Actin were calculated and compared between treatment groups. Data presented are representative of 3 independent experiments with 3 biological replica per group. A one-way ANOVA with Tukey's correction was used to calculate differences between the groups * p ≤ 0.05, ** p ≤ 0.002, *** p ≤ 0.0002, **** p < 0.0001.

AS101 and MS-275 had a synergistic effect on IL-10 upregulation, suggesting IL-10 self-regulates through class 1 HDAC activity (**Fig 1A**).

## IL-10 and class 1 HDACs suppress both MyD88-dependent and MyD88-independent pathways

Stimulation of TLR4 ultimately leads to the activation of NF-κB through MyD88 dependent and MyD88 independent (TRIF mediated) mechanisms and modulation of this signaling cascade is a target of significant interest in drug development [24]. Functionally, NF-κB is known to demonstrate variable effects on transcriptional activation in the developing inflammatory response and understanding the regulation of NF-κB target genes is a subject of intense investigation [25]. We demonstrated that IL-10 inhibition with AS101 enhanced transcription of all of MyD88-dependent pathway factors in alveolar macrophages with LPS stimulation, including MyD88, IRAK4, IRAK1, HSP90, TRAF6, and TAK1, corroborating IL-10s anti-inflammatory properties through suppression of the entire TLR-MyD88 axis (**Fig 2**). In contrast, inhibition of class 1 HDACs only leads to upregulation of IRAK4, HSP90, TRAF6 and TAK1, suggesting that only the transcription of these factors in the TLR4-MyD88 axis are class 1 HDAC-dependent. Inhibition of both IL-10 and class 1 HDACs unsurprisingly led to upregulation of the entire MyD88-dependent pathway (**Fig 2**). Similarly, both IL-10 and class 1

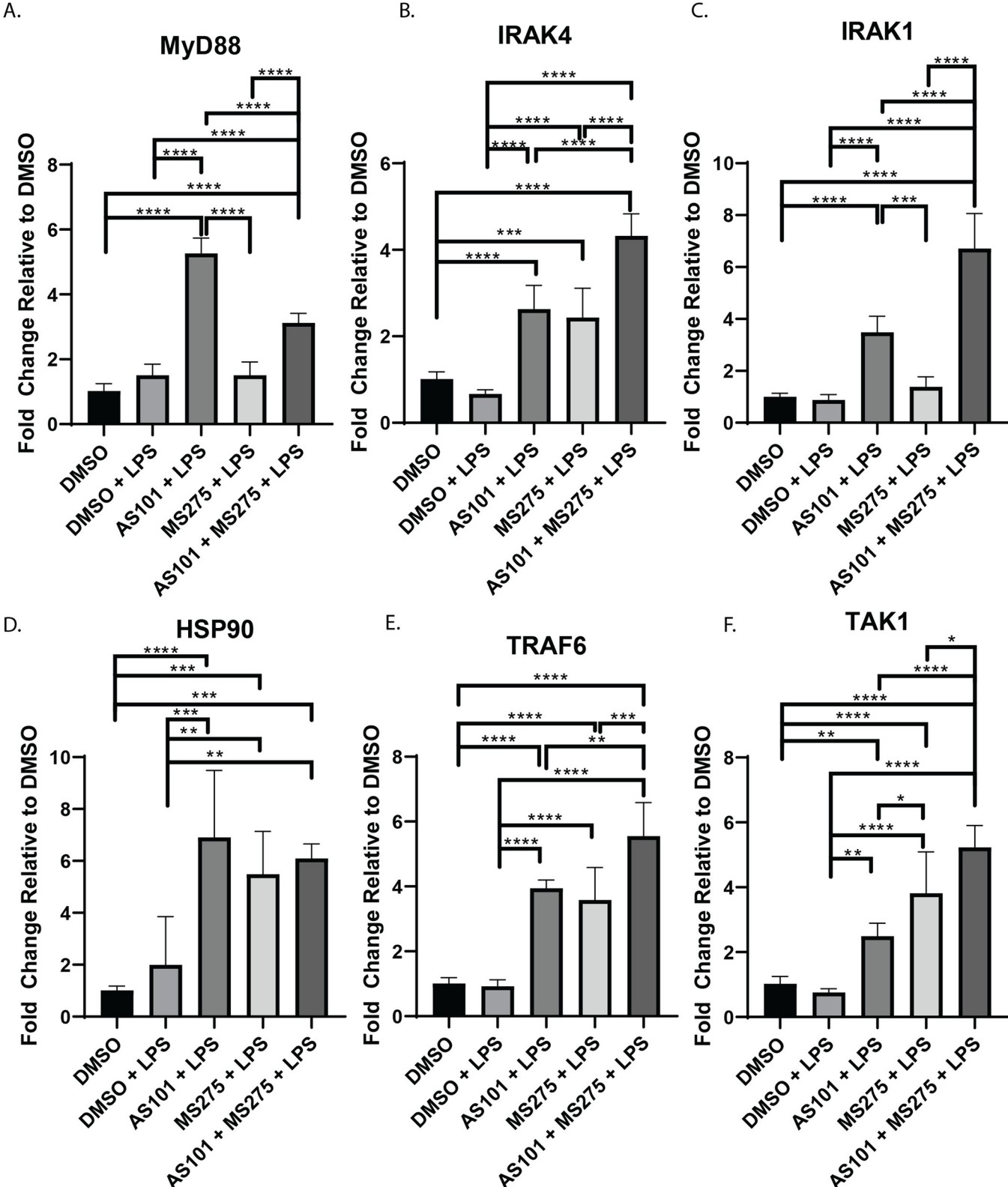

**Fig 2. Inhibition of class 1 HDACs regulates transcription of IRAK4 in MyD88-dependent TLR4 signaling.** Quantitative rtPCR analysis of RNA extracted from murine alveolar macrophage cells (MH-S) as previously described in Fig 1 and Materials and Methods. Relative quantities of A. MyD88, B. IRAK4, C. IRAK1, D. HSP90, E. TRAF6, and F. TAK1 were calculated and compared between treatment groups. Data presented are representative of 3 independent

experiments with 3 biological replica per group. A one-way ANOVA with Tukey's correction was used to calculate differences between the groups * p ≤ 0.05, ** p ≤ 0.002, *** p ≤ 0.0002, **** p < 0.0001.

HDAC inhibition independently and in combination upregulated both TRIF and RIP1 in LPS-stimulated alveolar macrophages, the key components of the MyD88-independent pathway (**Fig 3**).

## Inhibition of IL-10 and class 1 HDACs significantly decreases nuclear translocation of the p65 NF-κB subunit RelA via disruption of the phosphorylation/proteolytic degradation of IκBα in alveolar macrophages stimulated with LPS

Broad spectrum HDAC inhibition has been shown to inhibit NF-κB activation by suppressing the expression of key proteasome subunits which act to degrade IκBα, stabilizing the

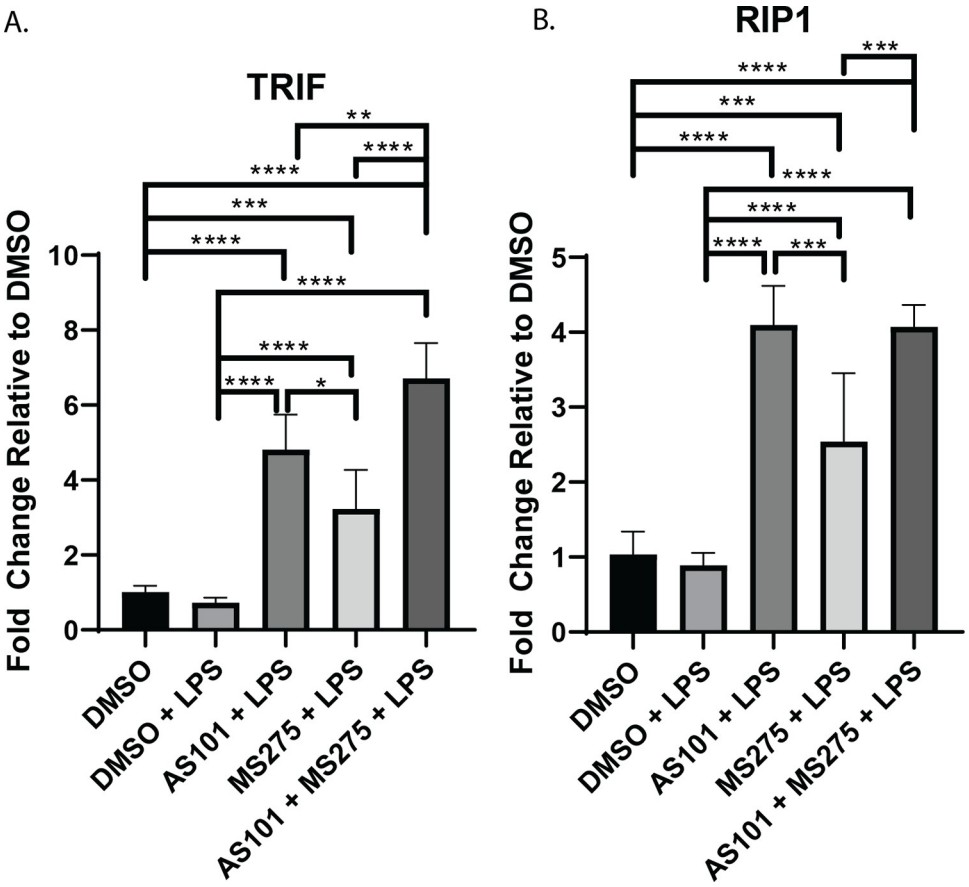

**Fig 3. Inhibition of IL-10 synthesis broadly affects transcription of TLR4 signaling components.** Quantitative rtPCR analysis of RNA extracted from murine alveolar macrophage cells (MH-S) as previously described in Figs 1 and 2, and Materials and Methods. Relative quantities of A. TRIF, and B. RIP1 were calculated and compared between treatment groups. Data presented are representative of 3 independent experiments with 3 biological replica per group. A one-way ANOVA with Tukey's correction was used to calculate differences between the groups * p ≤ 0.05, ** p ≤ 0.002, *** p ≤ 0.0002, **** p < 0.0001.

cytoplasmic NF-κB complex [26] A key step in the final activation of the NF-κB pathway is the degradation of IκBα, allowing NF-κB to translocate to the nucleus, where it activates proinflammatory gene expression, as discussed earlier. Here, we demonstrated that the AM cell line MH-S expresses high levels of phosphorylated IκBα at baseline, which is degraded with LPS stimulation. Only newly synthesized IκBα is unphosphorylated. AS101 enhances proteolytic degradation of phosphorylated IκBα and increases IκBα synthesis. However, treatment of these cells with a selective class 1 HDAC inhibitor (MS275) limited degradation of phosphorylated IκBα, and the combination of IL-10 and class 1 HDAC inhibition both interfered with the phosphorylation of IκBα and increased the synthesis of new unphosphorylated IκBα (**Fig 4A**). These findings suggest that one of the ways that IL-10 exerts its anti-inflammatory effects is through limiting IκBα phosphorylation. Similarly, class 1 HDAC inhibition confers its anti-inflammatory role at least through inhibition of IκBα degradation, limiting transport of NF-κB from the cytoplasm to the nucleus. Interestingly, inhibition of IL-10 synthesis increased transcription of IkKα, RelA, and the NF-κB p50 subunit, suggesting that a negative feedback loop exists to counter the anti-inflammatory effects of IL-10 during TLR activation (**Fig 4B–**

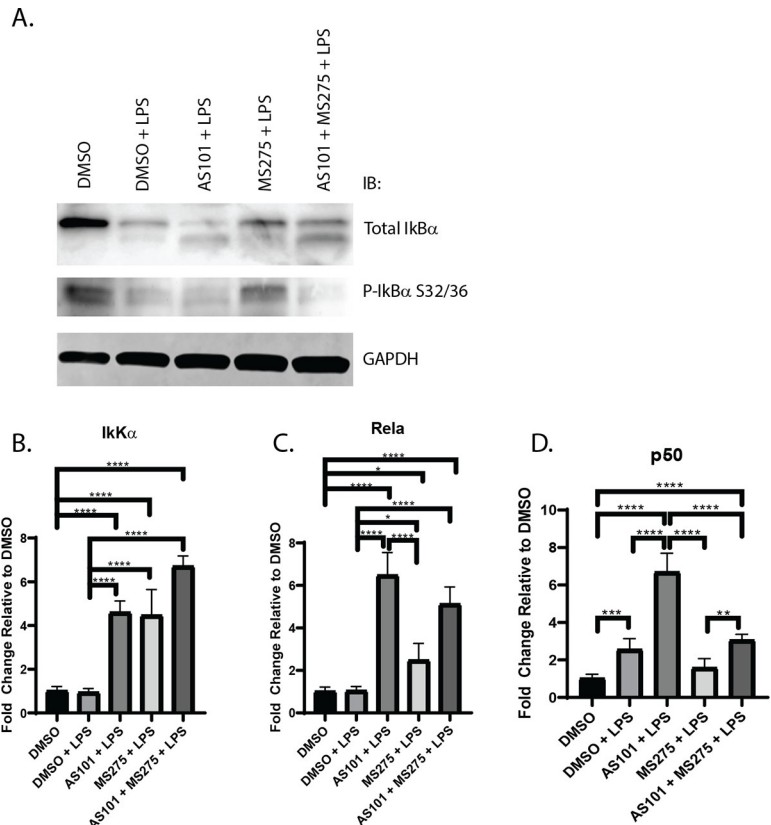

**Fig 4. Inhibition of IL-10 synthesis or class 1 HDACs differentially affect the transcription of core components of NFκB.** A. Western blot of total IκBα, phospho- IκBα and GAPDH in murine alveolar macrophage cells (MH-S). Visualized bands are at 39 kDa, 50 kDa and 36 kDa respectively. B-D. Quantitative PCR analysis of RNA extracted from murine alveolar macrophage cells (MH-S) treated with DMSO, treated with DMSO and stimulated with 100 ng/mL LPS (DMSO + LPS), treated with 300 nM AS101 and stimulated with 100 ng/mL LPS (AS101 + LPS), treated with 10 μM MS275 and stimulated with 100 ng/mL LPS (MS275 + LPS), and treated with both 300 nM AS101 and 10 μM MS275 and treated with 100 ng/mL LPS (AS101 + MS275 + LPS). Relative quantities of B. IkKα, C. Rela, and D. p50 were calculated and compared between treatment groups. Data presented are representative of 3 independent experiments with 3 biological replica per group. A one-way ANOVA with Tukey's correction was used to calculate differences between the groups * $p \leq 0.05$, ** $p \leq 0.002$, *** $p \leq 0.0002$, **** $p < 0.0001$.

**4D**). In contrast, class 1 HDACs inhibition only increases IkKα, and RelA, but not p50 (**Fig 4B–4D**). Confirming the effect on IκBα presented in Fig 4, disruption of the IL-10—class 1 HDAC signaling axis significantly decreased the nuclear translocation of RelA in alveolar macrophages stimulated with LPS (**Fig 5**). Of note, RelA was not shown to increase its nuclear density significantly between the DMSO and DMSO + LPS groups, likely due to the timing of microscopy, which took place 24 hours after stimulation. It has been shown that RelA exits the nucleus within 4 hours after stimulation [27].

## IL-10 is constitutively produced in AM, and inhibition of both IL-10 and class 1 HDACs upregulates IL-12b

Inhibition of class 1 HDACs, specifically HDAC3, has been shown to enhance the secretion of IL-12b in bone marrow derived macrophages in the presence of LPS [12]. The IL-10 protein synthesis inhibitor AS101 is known to interfere with IL-10, and also enhance TNFα secretion [14]. Similar mechanisms of action have been described for AS101 by its ability to specifically inhibit the secretion of IL-1β and IL-18 post-translationally [28]. Here, we demonstrate that IL-10 was constitutively synthesized in AM at baseline (~5 pg/mL in our culture conditions), is increased with LPS stimulation, and treatment with AS101 in the presence of LPS reversed this increase to base levels (**Fig 6A**). We also show that, not only does inhibition of IL-10 synthesis increased secretion of IL-12b, but inhibition of class 1 HDACs also increased secretion of IL-12b even further. Interestingly, the combination of IL-10 and class 1 HDAC inhibition in the presence of LPS did not synergistically enhance IL-12b secretion with levels comparable to the IL-10 inhibition in the presence of LPS alone (**Fig 6B**). As expected, AS101 stimulated TNFα secretion, however, class 1 HDAC inhibition had the opposite effect in the presence of LPS (**Fig 6C**). This, together with the finding that IL-10 is constitutively expressed in AM, suggests that IL-10 curbs IL-12b synthesis. Interestingly, the combination of IL-10 and class 1 HDAC inhibition demonstrates less secreted IL-12b in culture, suggesting that the IL-10 inhibitor plays a more complex role on cytokine secretion than previously identified (**Fig 6B**). Class 1 HDAC inhibition in the presence of LPS upregulated IL-10 secretion, and significantly decreased secretion of TNFα (**Fig 6A and 6C**). Of note, IL-10 mRNA transcription was noted to remain unchanged with LPS stimulation (**Fig 1A**), while protein synthesis increased (**Fig 6A**). It appears that anti-inflammatory IL-10 transcription takes place constitutively with an inhibitory process limiting protein secretion, and stimulation with LPS enables removes this inhibitory step to allow for greater protein synthesis. The topic of post-transcriptional IL-10 regulation is to be examined in a separate project. Although AS101 increased TNFα secretion, inhibition of class 1 HDACs appears to play a dominant negative role on TNFα secretion (**Fig 6C**).

## IL-10 and class 1 HDACs differentially effect the secretion of neutrophil chemoattractants CXCL2, IL-6 and MIF

Recruitment of leukocytes to the site of injury is a key component of the innate immune response. Specifically in infectious ALI, neutrophils are recruited to the lung parenchyma to assist with pathogen elimination in the acute setting [29]. This recruitment is largely orchestrated by cytokines IL-6 and CXCL2, both end products of the NF-κB pathway activation [29, 30]. Macrophage Migration Inhibitory Factor (MIF) is a pleiotropic cytokine with many roles in inflammation and disease [31]. Importantly MIF is known to modulate bacterial detection by TLR4 [32]. Macrophage MIF secretion is known to be dose-dependent after LPS stimulation [33], suggesting a more complex regulatory mechanism beyond NF-κB, that is beyond the scope of the current project. Here, we demonstrate that inhibition of IL-10 synthesis and class

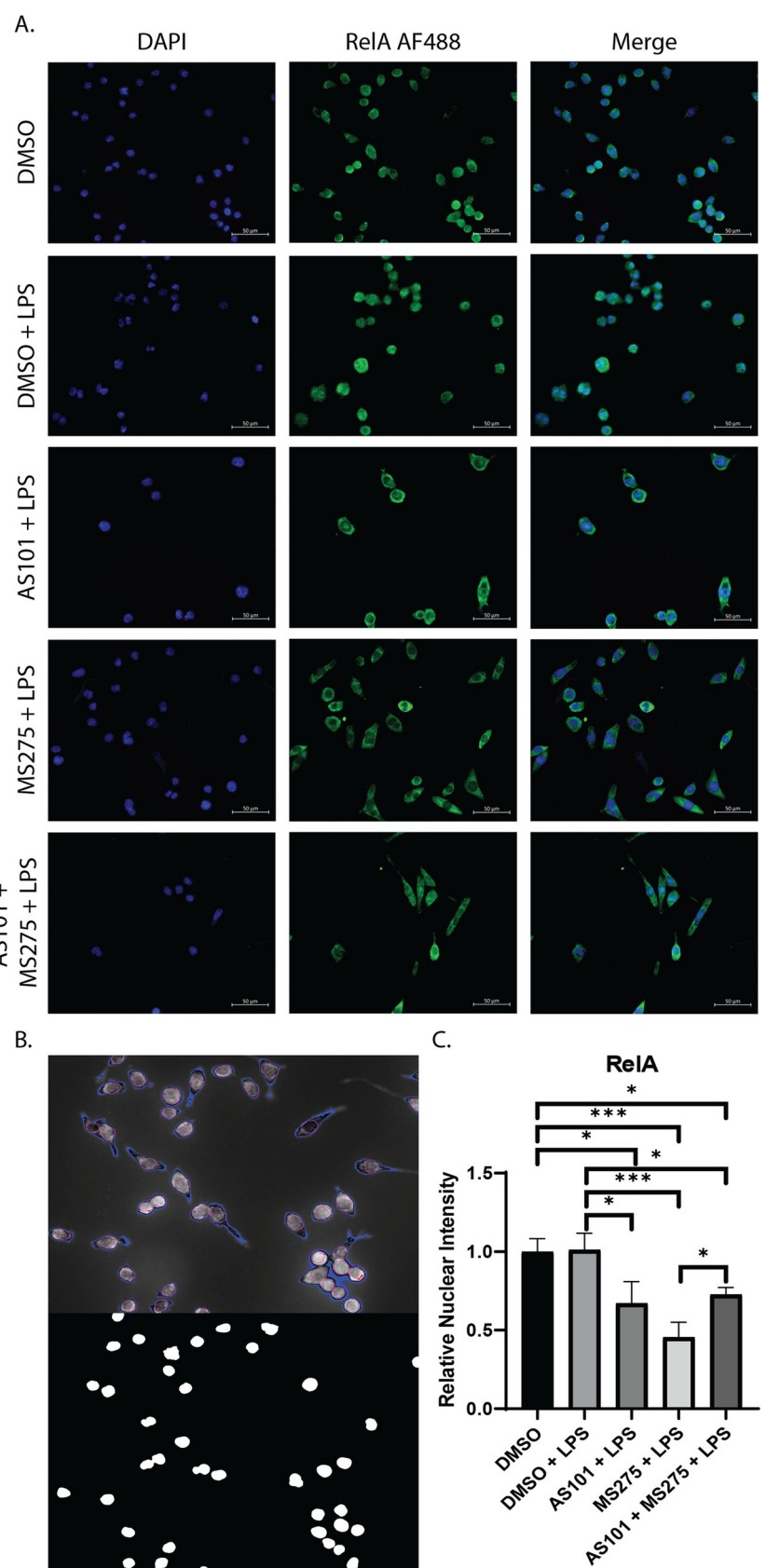

**Fig 5. Inhibition of class 1 HDACs reduces nuclear translocation of the p65 RelA NFκB subunit.** A. Immunofluorescence microscopy images of murine alveolar macrophage cells (MH-S) treated with either DMSO, DMSO and stimulated with 100 ng/mL LPS (DMSO + LPS), treated with 300 nM AS101 and stimulated with 100 ng/ mL LPS (AS101 + LPS), treated with 10 μM MS275 and stimulated with 100 ng/mL LPS (MS275 + LPS), and treated with both 300 nM AS101 and 10 μM MS275 and treated with 100 ng/mL LPS (AS101 + MS275 + LPS). Cells were then stained with mouse anti-RelA IgG and visualized under a 40x objective. Nuclear DNA was stained with DAPI (4′,6-diamidino-2-phenylindole). B. Representative image of the ImageJ Intensity Ratio Nuclei Cytoplasm Tool output for background correction (Top panel blue pixels) and nuclear location (Bottom panel). C. Relative nuclear intensity of RelA normalized to the DMSO control. Three random images were collected per treatment with at least 3 cells per field. A one-way ANOVA with Tukey's correction was used to calculate differences between the groups * $p \leq 0.05$, ** $p \leq 0.002$, *** $p \leq 0.0002$, **** $p < 0.0001$.

1 HDACs play pivotal roles in the secretion of these signaling molecules. LPS expectedly induced secretion of both CXCL2 and IL-6 in AM, but not MIF (**Fig 7**). Only class 1 HDAC, but not IL-10, inhibition downregulated CXCL2 secretion, while only IL-10, but not class 1 HDAC inhibition downregulated IL-6. Inhibition of both, interestingly reversed the inhibitory effect of class 1 HDACs on CXCL2, reversing it to inflammatory levels, while enhancing IL-6 synthesis (**Fig 7A and 7B**). MIF was secreted with both IL-10 and class 1 HDAC inhibition, and combination of both enhanced it even further (**Fig 7C**).

## Discussion

When pathogen associated molecular patterns (PAMPS) are detected by macrophages, commonly the first antigen presenting cell (APC) to initiate the immune response, they bind to Toll-Like Receptors. TLRs are a well-described family of type I transmembrane receptors commonly expressed on the cell surface and throughout the cytoplasm in endosomes of all cells, including AM. Recent work has described the activation of both TLR2 (recognition of bacterial cell wall components) and TLR4 (recognition of lipopolysaccharide) as a consequence of bacterial mediated sepsis leading to ARDS [34, 35]. However TLR4-dependent inflammatory responses have been shown to be essential to the development of ARDS in murine models [36], and in contrast to TLR2, only TLR4 is associated with alveolar macrophages activation in human lungs [37]. This interaction leads to intracellular signaling mediated chiefly through two adaptor protein systems, the Myeloid Differentiation Response Protein 88 (MyD88) and TIR-domain-containing adapter-inducing interferon-β (TRIF). TLR4 is unique among TLRs as it can signal via both MyD88 dependent and independent (TRIF dependent) pathways [38]. Activation of either pathway results in the phosphorylation and proteolytic degradation of IκBα enabling release of NF-κB. At the next step, NF-κB translocates into the nucleus to induce the expression of numerous genes that regulate the innate inflammatory response, including proinflammatory (TNFα, IL-1β, IL-6, IL-12b) and anti-inflammatory cytokines (IL-10), chemokines (CXCL-2, MIF), antimicrobial molecules (hydrolases, peptidases, and proteases, that lead to local tissue injury and, hence, ALI), along with MHC and co-stimulatory molecules required for adaptive immune activation [39]. This series of events enables the host to rapidly mount a defense against microbial invasion. The released chemokines attract neutrophils to the pulmonary parenchyma, which further release oxidants, nucleus acids and proteases worsening local cell necrosis [40]. Dead and dying cells release proteins and nucleic acids that act as powerful damage associated molecular patters (DAMPs) in their own right [41]. This inappropriate, ongoing recruitment of additional TLRs perpetuates the proinflammatory storm [40, 42, 43] and has been associated with several systemic metabolic and hemodynamic disturbances that can be more harmful to the host than the inciting trigger. Several clinical syndromes related to such a maladaptive innate immune response have been described, such as ALI/ARDS and sepsis. To minimize such potentially lethal occurrences, the immune system

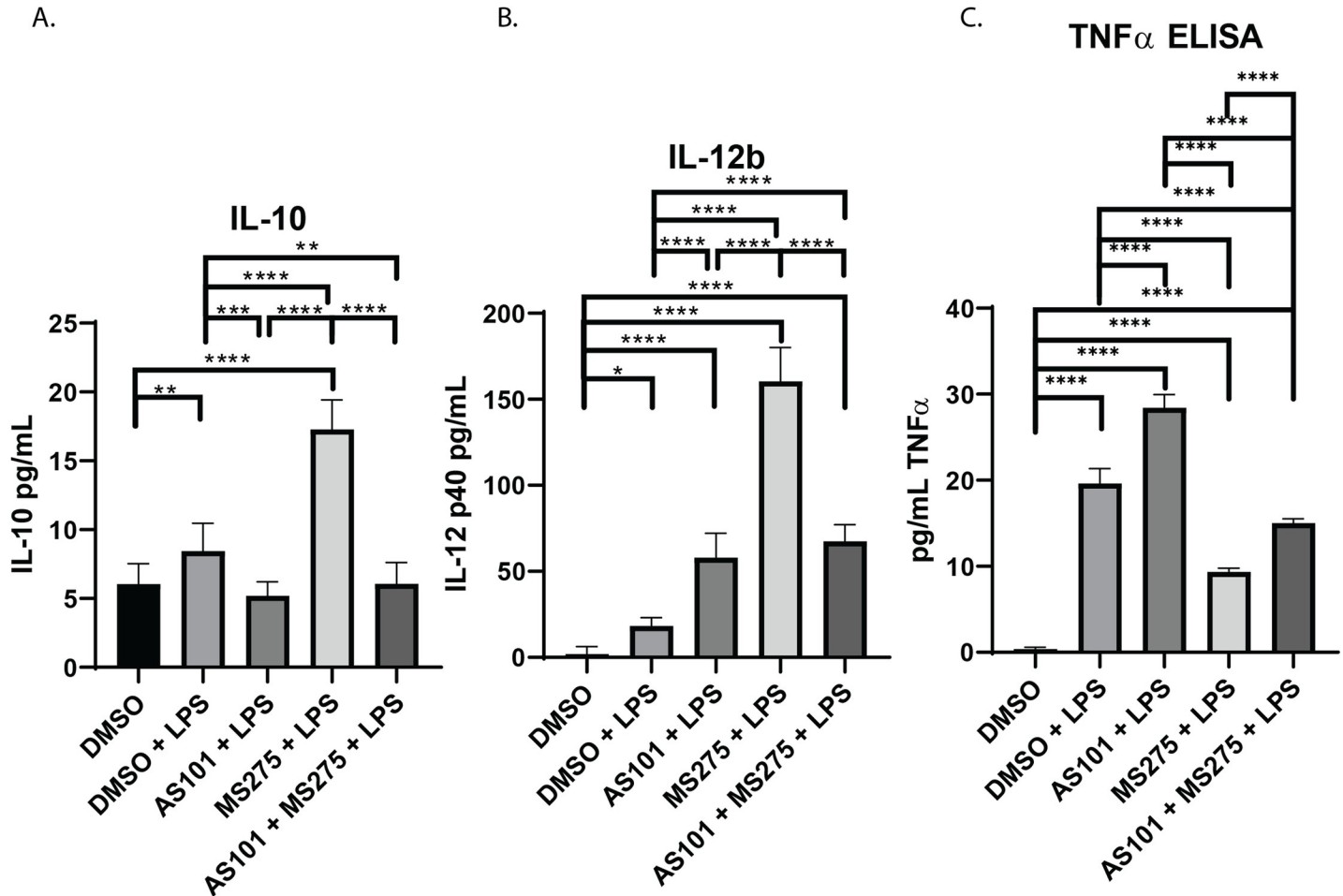

**Fig 6. Inhibition of IL-10 synthesis or class 1 HDACs has variable effects on IL-12b secretion.** Quantitative ELISA analysis of supernatant collected from murine alveolar macrophage cells (MH-S) cultures treated with DMSO, treated with DMSO and stimulated with 100 ng/mL LPS (DMSO + LPS), treated with 300 nM AS101 and stimulated with 100 ng/mL LPS (AS101 + LPS), treated with 10 μM MS275 and stimulated with 100 ng/mL LPS (MS275 + LPS), and treated with both 300 nM AS101 and 10 μM MS275 and treated with 100 ng/mL LPS (AS101 + MS275 + LPS). Precise quantities of A. IL-10, B. IL-12b, and C. TNFα (known to be stimulated by AS101) were measured and compared between treatment groups. Data presented are representative of 3 independent experiments with 3 biological replica per group. A one-way ANOVA with Tukey's correction was used to calculate differences between the groups * p ≤ 0.05, ** p ≤ 0.002, *** p ≤ 0.0002, **** p < 0.0001. Brackets represent individual comparisons. Line represents all groups were significantly different.

has evolved over time to include parallel anti-inflammatory mechanisms that act to curb the proinflammatory cascade, limit potential host tissue damage, and restore tissue homeostasis [44].

One of the key downstream proinflammatory cytokines that is newly synthesized from the TLR- NF-κB system activation is IL-12. It is an interleukin typically produced by myeloid and human B-cells in response to antigenic stimulation of the TLR system. It is a heterodimer, comprising of IL-12a (p35) and IL-12b (p40) subunits. The latter IL-12b is also a component of IL-23 (with a second p19 subunit). IL-27 (p28 and Ebi3 subunits) and IL-35 (p35 and Ebi3 subunits) are also included in a group of cytokines collectively known as the IL-12 cytokine family. All IL-12 family cytokines initiate intracellular signaling through various JAK-STAT pathways [45]. IL-12 is involved in the differentiation of T cells into Th1 cells [46] and enhances the cytotoxicity of Natural Killer cells and CD8+ cytotoxic T-cells, also through activation of the JAK-STAT pathway [47]. In addition to the bridging of the innate with the

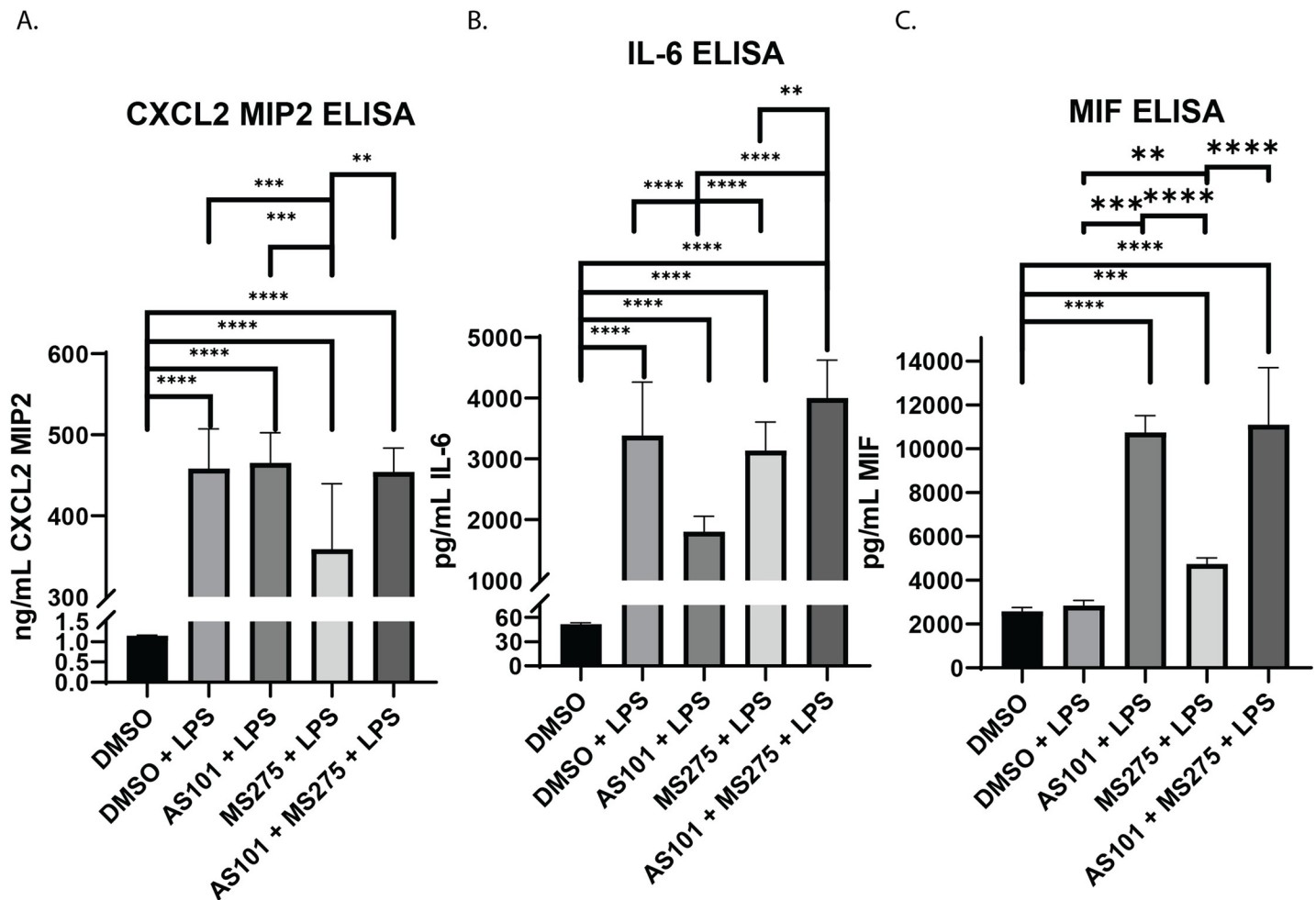

**Fig 7. Inhibition of IL-10 synthesis or class 1 HDACs have variable effects on the secretion of chemotactic cytokines and chemokines.** Quantitative ELISA analysis of supernatants collected from murine alveolar macrophage cells (MH-S) cultures treated with DMSO, treated with DMSO and stimulated with 100 ng/mL LPS (DMSO + LPS), treated with 300 nM AS101 and stimulated with 100 ng/mL LPS (AS101 + LPS), treated with 10 μM MS275 and stimulated with 100 ng/mL LPS (MS275 + LPS), and treated with both 300 nM AS101 and 10 μM MS275 and treated with 100 ng/mL LPS (AS101 + MS275 + LPS). Precise quantities of A. CXCL2 (MIP2), B. IL6, and C. MIF were measured and compared between treatment groups. Data presented are representative of 3 independent experiments with 3 biological replica per group. A one-way ANOVA with Tukey's correction was used to calculate differences between the groups $^*$ p ≤ 0.05, $^{**}$ p ≤ 0.002, $^{***}$ p ≤ 0.0002, $^{****}$ p < 0.0001.

adaptive immune response, IL-12 has also been found to regulate the APC function from which they emanated: Macrophages express functional IL-12 receptors that are upregulated following activation, which enhance antigen presentation [48].

On the other hand, IL-10 is a potent anti-inflammatory cytokine that plays a key role in limiting excess inflammation, countering the effects of IL-12 [49, 50]. It is produced mainly by monocytes, and to a lesser extent lymphocytes, by activation of TLR or Fc receptor pathways [51], with NF-κB being one of its most potent transcription factors [51]. IL-10 may autoregulate its expression via a negative feedback loop through autocrine stimulation of the IL-10 receptor [52], or post-transcriptionally [53–55], IL-10 confers several immune regulatory effects by downregulating the expression of Th1 cytokines, MHC class II antigens, and co-stimulatory molecules on macrophages; while enhancing B cell survival, proliferation, and antibody production. IL-10 also has been found to regulate the JAK-STAT signaling pathway (which controls IL-12b synthesis), and inhibits LPS and bacterial product-induced

transcription of pro-inflammatory cytokines TNFα [56], IL-1β [56], IL-12 [57], and IFNγ [58] secretion from TLR activation in myeloid lineage cells.

Kobayashi et al. [12] also demonstrated that IL-10 is necessary for intestinal macrophage tolerance, as wild-type intestinal macrophages produced IL-10, but not IL-12b, when stimulated with LPS, and this IL-10 synthesis was noted to be mediated through a MyD88-dependent pathway only [12]. Conversely, intestinal macrophages from IL-10$^{-/-}$ mice demonstrated robust IL-12b stimulation, and that IL-10 increased HDAC3 activity at the IL-12b promoter which led to histone deacetylation and transcriptional repression, suggesting that the immune homeostatic effects of IL-10 on IL-12b are mediated through HDAC3 (There were no differences in nucleosome remodeling, mRNA stability, NF-κB activation, or MAPK signaling to justify extended IL-12b transcription) [12]. Aste-Amezaga and colleagues similarly demonstrated IL-10-dependent suppression of both IL-12b and p35 gene transcription [57], though no p35 expression was detected in our system. Post translational acetylation of non-histone proteins cannot be ruled out as a potential contributing factor to changes in gene expression and protein synthesis. HDAC3 is known to interact with and actively deacetylate the NF-κB IκBα and p65 RelA subunits [59, 60]. Similarly, HDAC1 is known to interact with RelA and function to repress RelA target genes [61]. The changes in chromatin acetylation in combination with specific class 1 HDAC substrate acetylation could explain the changes in gene expression of NF-κB regulated genes observed in our study. The cytoplasmic and nuclear roles of HDACs need to be further defined to fully understand the immunomodulatory potential of pharmaceutically targeting HDACs or their substrates.

These observations may be applicable to intestinal macrophages that are constantly exposed to the intestinal microbiota and demonstrate immune tolerance, while imbalances between pro- and anti-inflammatory cytokine production in these APCs lead to low grade, chronic inflammation, unlike the explosive, immediate innate immune response noted when the otherwise sterile pulmonary environment is exposed to pathogens. In our experiments with AM that were cultured in sterile conditions, we demonstrate that IL-10 was also constitutively synthesized, regardless of prolonged pathogenic presence unlike intestinal macrophages [12]. Previous studies have demonstrated that AM isolated from healthy nonsmoking human volunteers activate the expression of IL-10 upon stimulation with LPS [62]. Additionally, loss of IL-10 has been attributed to significant morphological functional changes in aged mice [63]. This indicates that IL-10 is key mediator of IL-10 homeostasis in the murine lung with either low level constitutive or intermittent expression. This further supports our finding that IL-10 is expressed in unstimulated MH-S AM cells which similar to primary human AM increase the secretion of IL-10 when stimulated with LPS. The present study aimed to define the IL-10 class 1 HDAC signaling axis within AM and identify additional mediators of inflammation that may be targeted therapeutically.

We also demonstrate that IL-10 and class 1 HDACs both independently and synergistically regulate IL-12b transcription, and that IL-10 modulates the TLR -NF-κB signaling pathway by suppressing the transcription of both MyD88-dependent and MyD88-independent pathways. The latter finding provides an additional means of IL-10 modulating its own transcription and secretion (**Figs 1A and 6A**), in addition to regulating its own receptor [52], or post-transcriptionally [53–55], and corroborates the finding by He et al that TLR4-NF-κB signaling differentially regulates IL-10 and IL-12b transcription, through regulation of c-fos synthesis and differential binding of the NF-κB transcription factor to the two cytokine promoters. AS101 is known to disrupt IL-10 secretion by inhibiting integrin activity [64]. The loss of integrin activity likely explains the discrepancy observed between the transcriptional and secreted cytokine analysis. In addition, we showed that IL-10 and class 1 HDAC inhibition differentially affected IκBα synthesis and proteolytic degradation, shifting the TLR-NF-κB signaling to anti-

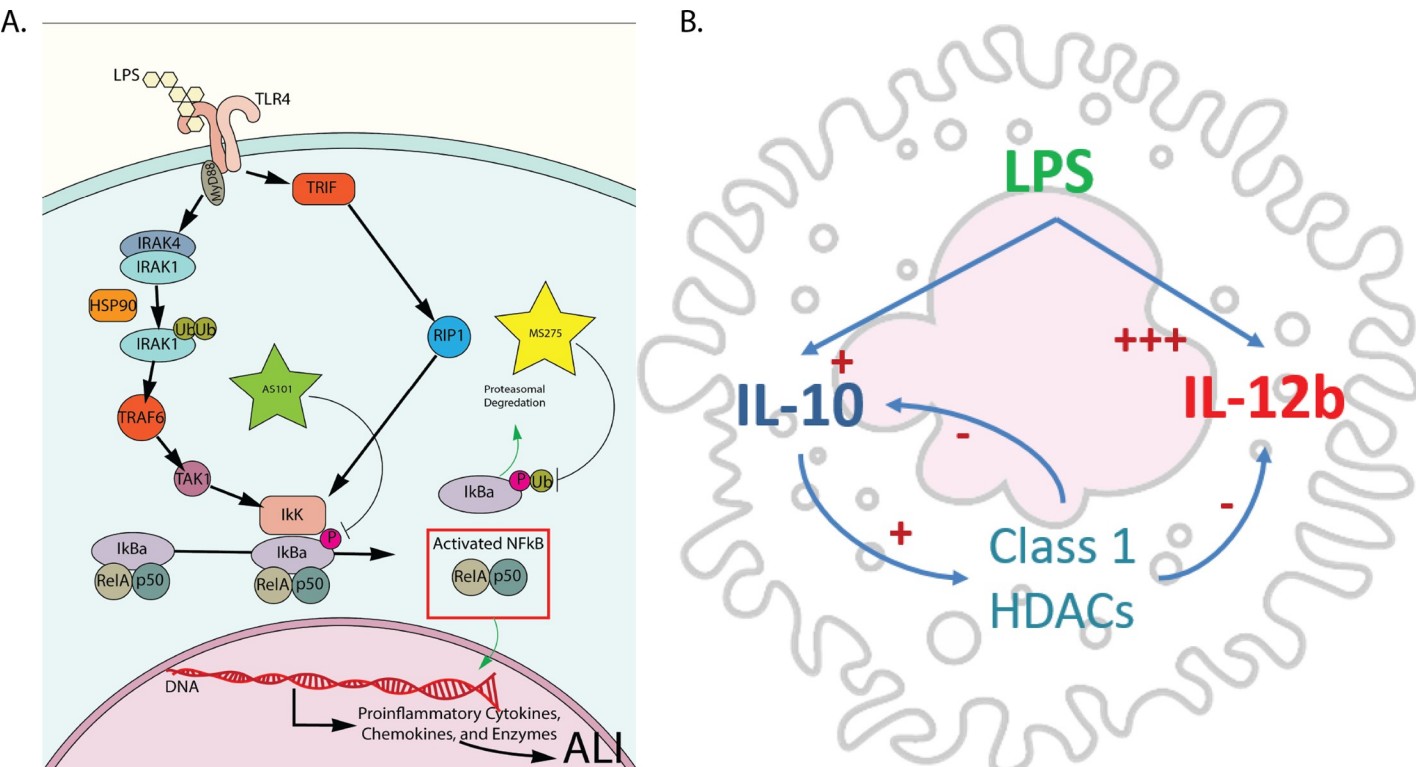

**Fig 8. Summary of pathways affected by inhibition of either IL-10 synthesis or class 1 HDACs.** A. Inhibition of IL-10 synthesis with AS101 broadly affects MyD88 dependent and independent TLR4 signaling via disruption of IκBα phosphorylation. Inhibition of class 1 HDACs specifically targets the ubiquitination/degradation of phosphorylated IκBα and reduces nuclear translocation of RelA/p50. B. Alveolar macrophage stimulated by LPS actively secrete IL-10 and IL-12b in response. IL-10 responses act to dampen proinflammatory IL-12b secretion and this process is mediated by IL-10 actively increasing class 1 HDACs activity to repress IL-12b transcription. Activated class 1 HDACs also acts to repress IL-10 secretion in an auto regulatory fashion.

inflammatory territory. Both enhance IkKα and RelA expression, but only IL-10 inhibition increases p50 gene expression. Lastly, we show that IL-10 inhibited the transcription of both MyD88-dependent and MyD88-independent pathway components, while class 1 HDAC inhibition only appears to affect the transcription of MyD88-dependent components. **Fig 8A and 8B** summarizes the effect of IL-10 and class 1 HDAC inhibition on the modulation of IκBα phosphorylation (AS101) and ubiquitination/degradation (MS275) in the TLR4 signaling pathway, and the proposed interaction between IL-10, IL-12b and class 1 HDACs with LPS stimulation. This work supports the idea that class 1 HDACs are significantly involved in the acute inflammatory response and can be important targets for specific immunomodulatory treatment strategies. Future work to address the role of specific class 1 HDACs (HDAC1, HDAC2, and or HDAC3) in the pathogenesis of ARDS in-vivo will involve the use of conditional knockout mice, as loss of these genes has been demonstrated to be embryonically lethal [65, 66]. Similarly, constitutive tissue-specific loss of these genes would lead to developmental abnormalities at best. Additionally, more specific pharmacological inhibition (HDAC1: Valproic Acid, HDAC2: Romidepsin, HDAC3: RGFP966) will be utilized to more specifically inhibit these enzymes.

We did not specifically examine the way(s) class 1 HDACs interact with IL-10 and IL-12b, nor the role specific class 1 HDACs play in these phenotypes as inhibition with MS-275 targets HDACs 1, 2, and 3. This constitutes the topic of a separate project we are currently pursuing and on which we will be reporting soon.

## Supporting information

**S1 File. Raw blots.**
(ZIP)

## Author Contributions

**Conceptualization:** Brent A. Stanfield, George Kasotakis.

**Data curation:** Brent A. Stanfield, George Kasotakis.

**Formal analysis:** Brent A. Stanfield, George Kasotakis.

**Funding acquisition:** George Kasotakis.

**Investigation:** Brent A. Stanfield.

**Methodology:** Brent A. Stanfield.

**Project administration:** Brent A. Stanfield, Todd Purves, Scott Palmer, Bruce Sullenger, Karen Welty-Wolf, Krista Haines, Suresh Agarwal.

**Resources:** Todd Purves.

**Software:** Brent A. Stanfield.

**Supervision:** Todd Purves, Scott Palmer, Bruce Sullenger, Karen Welty-Wolf, Krista Haines, Suresh Agarwal.

**Validation:** Brent A. Stanfield.

**Visualization:** Brent A. Stanfield.

**Writing – original draft:** Brent A. Stanfield, Todd Purves, Scott Palmer, Bruce Sullenger, Karen Welty-Wolf, Krista Haines, Suresh Agarwal, George Kasotakis.

**Writing – review & editing:** Brent A. Stanfield, Todd Purves, Scott Palmer, Bruce Sullenger, Karen Welty-Wolf, Krista Haines, Suresh Agarwal, George Kasotakis.

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
