## [Decision Letter · Decision Letter 0]

26 Aug 2020

PONE-D-20-22628

IL-10 AND HDAC3-DEPENDENT REGULATION OF TLR SIGNALING AND IL-12b SYNTHESIS IN ALVEOLAR MACROPHAGES

PLOS ONE

Dear Dr. Kasotakis,

Thank you for submitting your manuscript to PLOS ONE. After careful consideration, we feel that it has merit but does not fully meet PLOS ONE’s publication criteria as it currently stands. Therefore, we invite you to submit a revised version of the manuscript that addresses the points raised during the review process.

Your manuscript was reviewed by two experts and both of them provided major feedbacks, which are provided below.

We look forward to receiving your revised manuscript.

Kind regards,

Partha Mukhopadhyay, Ph.D.

Academic Editor

PLOS ONE

Journal Requirements:

Reviewers' comments:

Reviewer's Responses to Questions

**Comments to the Author**

1. Is the manuscript technically sound, and do the data support the conclusions?

Reviewer #1: Partly

Reviewer #2: Partly

2. Has the statistical analysis been performed appropriately and rigorously? 

Reviewer #1: Yes

Reviewer #2: Yes

3. Have the authors made all data underlying the findings in their manuscript fully available?

Reviewer #1: Yes

Reviewer #2: Yes

4. Is the manuscript presented in an intelligible fashion and written in standard English?

Reviewer #1: Yes

Reviewer #2: Yes

5. Review Comments to the Author

Reviewer #1: In the present study "IL-10 AND HDAC3-DEPENDENT REGULATION OF TLR SIGNALING AND IL-12b SYNTHESIS IN ALVEOLAR MACROPHAGES" the authors try to discover the effect of IL-10 and HDAC3 on LPS induced alveolar macrophage inflammatory response. Here are some major concerns:

1. The current research utilizes MS-275 to discuss the function of HDAC3. However, it is an inhibitor of both HDAC1 and HDAC3. If the authors insist on claiming the effect of HDAC3, a specific inhibitor of HDAC3, or genetic modification of HDAC3 expression should be used. Judging from the current data, the authors can only make conclusions about the function of HDAC1 and HDAC3, instead of a specific HDAC3 function as claimed in the manuscript.

2. Western blot in Figure 3 should show both the phosphorylated band and the total protein band. A loading control should be shown as well.

3. The authors need to analyze the nuclear translocation of p65 and p50 to claim a regulation of NF-kappaB. Phosphorylation of IkappaBs and the expression level of signal molecules of this pathway are not sufficient to indicate its activation.

4. There is not any evidence about the upper/lower stream relationship of IL-10 and HDAC3. Because AS101 and MS-275 have combinatorial effect in many results, it is more likely IL-10 and HDAC3 go through independent mechanisms. The authors need to correct the manuscript and Figure 7.

5. In figure 1A, IL-12b has the highest mRNA expression when LPS is combined with AS-101 and MS-275. However, in the same condition IL-12b supernatant concentration is quite low as shown in Figure 5B. The authors need to explain the discrepancy.

6. In figure 1C, why would AS-101 increase the mRNA expression of HDAC3? Does it mean IL-10 inhibits the expression of HDAC3, which is against the authors’ theory?

7. The authors need to discuss why TNF-alpha, MIF, IL-6 and CXCL2 show different pattern than IL-12b, as they are supposed to be regulated by NF-kappaB too.

8. The authors need to discuss why alveolar macrophages show different pattern than other macrophages in their cytokine/chemokine expression upon stimulation with LPS, especially the lack of IL-10 induction.

Reviewer #2: The manuscript entitled “IL-10 and HDAC3 dependent regulation of TLR signaling and IL-12b synthesis in alveolar macrophages” demonstrated that alveolar macrophage (AM) upregulates IL-10 and IL-12b production with a high HDAC3 dependent manner. Therefore, the authors claimed that HDAC3 might be a potential target for macrophage-initiated pulmonary inflammation in acute lung injury. Throughout this article, the authors clearly depicted the balance between pro-inflammatory and anti-inflammatory cytokine production after LPS treatment in vitro. In addition, the authors utilized inhibitions of IL-10 and HDAC to alveolar macrophage to support the hypothesis. Currently, the acute respiratory distress syndrome (ARDS) has become an important clinical challenge and we need to understand the pathogenesis of this disease. The article will be helpful to understand the underlying mechanisms of ARDS. However, several concerns must be addressed for better quality.

1) Acute Respiratory Distress syndrome (ARDS) can be induced by a variety immunological responses and the activation of TLR. According to previous reports (S Han and RK Mallampalli, 2015), ARDS induces both TLR2 (microbial cell wall components) and TLR4 (LPS) as a consequence of bacterial mediated sepsis. The authors only focused on TLR4 in this article. Please provide more rationale to choose TLR4 and support idea the involvement of TLR2 in Acute lung injury.

2) The authors used alveolar macrophage cells (MH-S) in this study. However, it is very difficult to make a conclusion only with in vitro system. Although bacterial pneumonia is the most common cause of ARDS, there are a variety of intercellular communications to induce ARDS. It is unlikely to demonstrate the pathogenesis of ARDS via only the involvement of alveolar macrophage cells. Please consider using animal model for in-depth understanding. For example, LPS injection to the animal model to mimic ARDS with AS101/MS275 treatment.

3) Related with previous suggestion, using IL-10 KO mouse or shIL-10 Knock down mice would be recommended to understand this pathogenesis.

4) There was no data about TLR4 activity or TLR4 changes with LPS stimulation. It should be added.

5) In line139, “AS101 inhibited IL-10 protein synthesis”, however, there is no AS101 treated group data to compare with vehicle treated group. Once again, there were no data “AS-101 only treated” and “MS275 only treated” group in most graphs. Please add those data, too. For instance, down-regulation of IL-10/HDAC3 mRNA level must be pre-qualified after inhibitory chemical treatment (whether IL-10 inhibitor and HDAC3 inhibitor work well or not).

6) Please put the statistics between the DMSO and DMSO+LPS groups in figures.

7) In figure legends, there are many repeats in each legend. Please make them more simplify.

8) In Figure 4, there is no house keeping protein in Western blot results.

9) In Figure 5c and Figure 6B (AS101+LPS), Please re-identify the statistical analyses. It is confusing.

6. PLOS authors have the option to publish the peer review history of their article (what does this mean?). If published, this will include your full peer review and any attached files.

Reviewer #1: No

Reviewer #2: No

---

## [Author Response · Author response to Decision Letter 0]

19 Oct 2020

We want to thank the PLOS One Editorial Team for the opportunity to address the issues identified in the peer-review process, and the reviewers for their insightful, constructive feedback. We do believe that the edits made based on the reviewers’ suggestions significantly enhance the quality and impact of our manuscript. 

The specific edits made are shown below each review item in red, while corresponding edits in the manuscript can be followed by Tracked Changes. 

Reviewer #1: In the present study "IL-10 AND HDAC3-DEPENDENT REGULATION OF TLR SIGNALING AND IL-12b SYNTHESIS IN ALVEOLAR MACROPHAGES" the authors try to discover the effect of IL-10 and HDAC3 on LPS induced alveolar macrophage inflammatory response. Here are some major concerns:

1. The current research utilizes MS-275 to discuss the function of HDAC3. However, it is an inhibitor of both HDAC1 and HDAC3. If the authors insist on claiming the effect of HDAC3, a specific inhibitor of HDAC3, or genetic modification of HDAC3 expression should be used. Judging from the current data, the authors can only make conclusions about the function of HDAC1 and HDAC3, instead of a specific HDAC3 function as claimed in the manuscript.

We appreciate the reviewer’s insightful note. We have edited the manuscript throughout to reflect that MS275 may inhibit all class 1 HDACs (HDAC1, HDAC2, and HDAC3). 

2. Western blot in Figure 3 should show both the phosphorylated band and the total protein band. A loading control should be shown as well.

We thank the reviewer for the comment and have added the S32/36 P-IκBα and GAPDH blots to the figure (Figure 4A)

3. The authors need to analyze the nuclear translocation of p65 and p50 to claim a regulation of NF-kappaB. Phosphorylation of IkappaBs and the expression level of signal molecules of this pathway are not sufficient to indicate its activation.

We thank the reviewer for the comment, and have added Figure 5 to the manuscript detailing the change in nuclear RelA with IL-10 and Class 1 HDAC inhibition under stimulation with LPS.

4. There is not any evidence about the upper/lower stream relationship of IL-10 and HDAC3. Because AS101 and MS-275 have combinatorial effect in many results, it is more likely IL-10 and HDAC3 go through independent mechanisms. The authors need to correct the manuscript and Figure 7.

We thank the reviewer for the comment and have changed Figure 7 (now Figure 8a) to specifically define AS101 as inhibiting IκBα phosphorylation, while MS275 specifically targets P-IκBα ubiquitination/degradation.

5. In figure 1A, IL-12b has the highest mRNA expression when LPS is combined with AS-101 and MS-275. However, in the same condition IL-12b supernatant concentration is quite low as shown in Figure 5B. The authors need to explain the discrepancy.

We thank the reviewer for the note. Based on this, we added an explanation of the mechanism of action AS101 has on cytokine secretion, based on previously published data. “AS101 is known to disrupt IL-10 secretion by inhibiting integrin activity (1). The loss of integrin activity likely explains the discrepancy observed between the transcriptional and secreted cytokine analysis.”

6. In figure 1C, why would AS-101 increase the mRNA expression of HDAC3? Does it mean 

We thank the reviewer for the comment, and have removed the HDAC3 expression analysis from the paper. MS275 inhibits class 1 HDACs (HDAC1, HDAC2, and HDAC3) independent of their transcriptional regulation. Disruption of class 1 HDAC function has broad effects on the transcriptional regulation of many genes, as evident by figures 1-4, and the mechanism of action of MS275 had been described in the first paragraph of the Methods section: “Agent MS275, a selective class 1 HDAC inhibitor (2) (Entinostat, Santa Cruz Biotechnology, sc-279455), was reconstituted to a stock concentration of 10 mM in DMSO.”

7. The authors need to discuss why TNF-alpha, MIF, IL-6 and CXCL2 show different pattern than IL-12b, as they are supposed to be regulated by NF-kappaB too.

We thank the reviewer for the comment and acknowledge the importance of NF-κB in the expression of TNF-alpha, MIF, IL-6, CXCL2, and IL-12b. As a potent inducer of nuclear NF-κB, LPS did induce the secretion of TNF-alpha, IL-6, CXCL2, IL-10, and IL-12b over base line concentrations, however the mechanism by which IL-10 and class 1 HDACs modulate the secretion of these cytokines is a topic to be investigated in future work. Macrophage MIF secretion is known to be dose-dependent after LPS stimulation, (3) suggesting a more complex regulatory mechanism beyond merely NF-κB, and this regulation is beyond the scope of the current project. The following has been added to the Results section, paragraph 10: “Macrophage MIF secretion is known to be dose-dependent after LPS stimulation, (3) suggesting a more complex regulatory mechanism beyond NF-κB, that is beyond the scope of the current project.”

8. The authors need to discuss why alveolar macrophages show different pattern than other macrophages in their cytokine/chemokine expression upon stimulation with LPS, especially the lack of IL-10 induction.

We thank the reviewer for the comment, and apologize for the confusion. LPS did induce IL-10 secretion over baseline levels. To better show this, we have added the statistical comparisons between the vehicle (DMSO) treatment group and the other 4 groups in the study (Figure 6A).

Reviewer #2: The manuscript entitled “IL-10 and HDAC3 dependent regulation of TLR signaling and IL-12b synthesis in alveolar macrophages” demonstrated that alveolar macrophage (AM) upregulates IL-10 and IL-12b production with a high HDAC3 dependent manner. Therefore, the authors claimed that HDAC3 might be a potential target for macrophage-initiated pulmonary inflammation in acute lung injury. Throughout this article, the authors clearly depicted the balance between pro-inflammatory and anti-inflammatory cytokine production after LPS treatment in vitro. In addition, the authors utilized inhibitions of IL-10 and HDAC to alveolar macrophage to support the hypothesis. Currently, the acute respiratory distress syndrome (ARDS) has become an important clinical challenge and we need to understand the pathogenesis of this disease. The article will be helpful to understand the underlying mechanisms of ARDS. However, several concerns must be addressed for better quality. 

1. Acute Respiratory Distress syndrome (ARDS) can be induced by a variety immunological responses and the activation of TLR. According to previous reports (S Han and RK Mallampalli, 2015), ARDS induces both TLR2 (microbial cell wall components) and TLR4 (LPS) as a consequence of bacterial mediated sepsis. The authors only focused on TLR4 in this article. Please provide more rationale to choose TLR4 and support idea the involvement of TLR2 in Acute lung injury.

We thank the reviewer for the comment and have added the above mentioned reference to the manuscript. Also, we went on the describe our rational for choosing TLR4 over TLR2 in our investigation by including references describing the importance of TLR4 activation to the development of ARDS and the dual signaling mechanism (MyD88 Dependent and Independent) nature of TLR4 unique to TLR4. The following has been added to the first paragraph of the Discussion section: “Recent work has described the activation of both TLR2 (recognition of bacterial cell wall components) and TLR4 (recognition of lipopolysaccharide) as a consequence of bacterial mediated sepsis leading to Acute Respiratory Distress Syndrome (ARDS) (4, 5). However TLR4 dependent inflammatory responses have been shown to be essential to the development of ARDS in murine models (6) and in contrast to TLR2, only activation of TLR4 is associated with the activation of alveolar macrophages in human lungs (7). This interaction leads to intracellular signaling mediated chiefly through two adaptor protein systems, the Myeloid Differentiation Response Protein 88 (MyD88) and TIR-domain-containing adapter-inducing interferon-β (TRIF). TLR4 is unique among TLRs as it can signal via both MyD88 dependent and independent (TRIF dependent) pathways (8).”

2. The authors used alveolar macrophage cells (MH-S) in this study. However, it is very difficult to make a conclusion only with in vitro system. Although bacterial pneumonia is the most common cause of ARDS, there are a variety of intercellular communications to induce ARDS. It is unlikely to demonstrate the pathogenesis of ARDS via only the involvement of alveolar macrophage cells. Please consider using animal model for in-depth understanding. For example, LPS injection to the animal model to mimic ARDS with AS101/MS275 treatment.

We thank the reviewer for the comment, and indeed plan to conduct detailed animal studies on the role of class 1 HDACs in animal models of ARDS. Animal experiments are beyond the scope of the current project. In addition, generation of knockout mice, as requested, was not feasible within the timeframe of the response to review, given our ability to only run our lab part time, due to institutional Covid-related restrictions. We include a detailed description of the planned animal work in paragraph 6 of the Discussion section: “Future work to address the role of specific class 1 HDACs (HDAC1, HDAC2, and or HDAC3) in the pathogenesis of ARDS in-vivo will involve the use of conditional knockout mice, as loss of these genes has been demonstrated to be embryonically lethal (9, 10). Similarly, constitutive tissue-specific loss of these genes would lead to developmental abnormalities at best. Additionally, more specific pharmacological inhibition (HDAC1: Valproic Acid, HDAC2: Romidepsin, HDAC3: RGFP966) will be utilized to more specifically inhibit these enzymes.”

3. Related with previous suggestion, using IL-10 KO mouse or shIL-10 Knock down mice would be recommended to understand this pathogenesis.

We thank the reviewer for the suggestion. Please see response to item #3. We do plan on integrating IL-10 knockout mice on our upcoming experiments.

4. There was no data about TLR4 activity or TLR4 changes with LPS stimulation. It should be added.

We thank the reviewer for the comment. TLR4 activation with LPS stimulation in macrophages is a well described phenomenon that has been described extensively in the literature. The following has been added to the 1st paragraph of the Results section: “LPS stimulation has been extensively shown to rapidly induce TLR4 activation in macrophages (11-14) and previous work has described the downstream regulatory role IL-10 plays on the expression of IL-12b through the activation of HDAC3 (a class 1 HDAC) in colonic macrophages (15).”

5. In line139, “AS101 inhibited IL-10 protein synthesis”, however, there is no AS101 treated group data to compare with vehicle treated group. Once again, there were no data “AS-101 only treated” and “MS275 only treated” group in most graphs. Please add those data, too. For instance, down-regulation of IL-10/HDAC3 mRNA level must be pre-qualified after inhibitory chemical treatment (whether IL-10 inhibitor and HDAC3 inhibitor work well or not).

We thank the reviewer for the comment and have edited the manuscript throughout to reflect that the changes observed with AS101 and MS275 treatment are merely in the context of LPS stimulation.

6. Please put the statistics between the DMSO and DMSO+LPS groups in figures.

We thank the reviewer for the suggestion, and based on it, we have added the requested statistics to all the figures.

7. In figure legends, there are many repeats in each legend. Please make them more simplify.

We thank the reviewer for the comment. We have simplified the legends across all Figures.

8. In Figure 4, there is no house-keeping protein in Western blot results.

We thank the reviewer for the comment. We added the S32/36 P- IκBα and GAPDH blots to the figure (Figure 4A). 

9. In Figure 5c and Figure 6B (AS101+LPS), Please re-identify the statistical analyses. It is confusing.

We apologize for the confusion, and thank the reviewer for the suggestion. We re-identified the statistical analysis in the requested graphs, now renamed Figure 6C and 7B.

---

## [Decision Letter · Decision Letter 1]

9 Nov 2020

PONE-D-20-22628R1

IL-10 AND CLASS 1 HISTONE DEACETYLASES ACT SYNERGISTICALLY AND INDEPENDENTLY ON THE SECRETION OF PROINFLAMMATORY MEDIATORS IN ALVEOLAR MACROPHAGES

PLOS ONE

Dear Dr. Kasotakis,

Thank you for submitting your manuscript to PLOS ONE. After careful consideration, we feel that it has merit but does not fully meet PLOS ONE’s publication criteria as it currently stands. Therefore, we invite you to submit a revised version of the manuscript that addresses the points raised during the review process.

Your manuscript was reviewed by the same reviewers and one of them provided few minor comments. Please address those comments.

We look forward to receiving your revised manuscript.

Kind regards,

Partha Mukhopadhyay, Ph.D.

Academic Editor

PLOS ONE

Reviewers' comments:

Reviewer's Responses to Questions

**Comments to the Author**

1. If the authors have adequately addressed your comments raised in a previous round of review and you feel that this manuscript is now acceptable for publication, you may indicate that here to bypass the “Comments to the Author” section, enter your conflict of interest statement in the “Confidential to Editor” section, and submit your "Accept" recommendation.

Reviewer #1: (No Response)

Reviewer #2: All comments have been addressed

2. Is the manuscript technically sound, and do the data support the conclusions?

Reviewer #1: Yes

Reviewer #2: (No Response)

3. Has the statistical analysis been performed appropriately and rigorously? 

Reviewer #1: Yes

Reviewer #2: (No Response)

4. Have the authors made all data underlying the findings in their manuscript fully available?

Reviewer #1: Yes

Reviewer #2: (No Response)

5. Is the manuscript presented in an intelligible fashion and written in standard English?

Reviewer #1: Yes

Reviewer #2: (No Response)

6. Review Comments to the Author

Reviewer #1: The current study "IL-10 AND CLASS 1 HISTONE DEACETYLASES ACT SYNERGISTICALLY AND INDEPENDENTLY ON THE SECRETION OF PROINFLAMMATORY MEDIATORS IN ALVEOLAR MACROPHAGES" made a lot of improvement compared with the previous version. Here are some more minor concerns to be addressed:

1. In figure 5, the DMSO + LPS group did not show increase of RelA nuclear localization compared with DMSO group. This group sets the baseline of the current experiment. If the authors believe it is because of a too late time point, an earlier time point should be selected for this experiment.

2. The authors claim AS101 reduces phosphorylation of IkBa. In such case, the subsequent degradation should be low and the total protein of IkBa should be high. In figure 4, the IkBa total protein is low in the AS101 group. Please explain. Also, how is this IL-10 synthesis inhibitor related to IkBa phosphorylation?

3. The authors addressed MYD88 dependent and independent pathways. However, the current study only detected the expression of the components of these pathways, which is not sufficient to indicate activity. The authors may do some discussions according to the current data, but putting these pathways in the conclusion is too stretchy.

4. The authors need to explain IL-10 mRNA expression is not increased by LPS in figure 1A but protein is increased in figure 6A.

Reviewer #2: (No Response)

7. PLOS authors have the option to publish the peer review history of their article (what does this mean?). If published, this will include your full peer review and any attached files.

Reviewer #1: No

Reviewer #2: No

---

## [Author Response · Author response to Decision Letter 1]

12 Dec 2020

Dear Reviewer, 

We appreciate the Reviewer’s additional insightful comments and constructive feedback, along with the opportunity to further edit our manuscript. 

Please see our response below. 

Reviewer #1: The current study "IL-10 AND CLASS 1 HISTONE DEACETYLASES ACT SYNERGISTICALLY AND INDEPENDENTLY ON THE SECRETION OF PROINFLAMMATORY MEDIATORS IN ALVEOLAR MACROPHAGES" made a lot of improvement compared with the previous version. Here are some more minor concerns to be addressed:

1. In figure 5, the DMSO + LPS group did not show increase of RelA nuclear localization compared with DMSO group. This group sets the baseline of the current experiment. If the authors believe it is because of a too late time point, an earlier time point should be selected for this experiment.

As our western blot demonstrated (Figure 4), MH-S cells appear to maintain a relatively high level of Phospho-IκBα in the unstimulated control group. Though IκBα is not readily degraded with this treatment, constitutive phosphorylation of IκBα could result in a steady state of nuclear NFκB activity difficult to overcome with LPS treatment. It is also possible that at the time point we investigated these treatments RelA was already exported to the cytoplasm. However, our key finding and focus here is that treatment with either MS275 or AS101 significantly reduced nuclear RelA localization. 

2. The authors claim AS101 reduces phosphorylation of IkBa. In such case, the subsequent degradation should be low and the total protein of IkBa should be high. In figure 4, the IkBa total protein is low in the AS101 group. Please explain. Also, how is this IL-10 synthesis inhibitor related to IkBa phosphorylation?

AS101 was applied at the time of LPS stimulation and may not have had time to interfere with the degradation of IκBα. Since Phospho- IκBα is constitutively high in these cells additional kinases (Other than IkKs) may be involved in this phenotype. This was not focus of this investigation but is definitely an interesting topic we plan to pursue in future work. In the discussion of figure 4 we state that it is likely that only newly synthesized IκBα is unphosphorylated and with constitutively high IκBα phosphorylation in MH-S cells additional regulatory steps are likely what contributes to the inhibition of IκBα degradation.

3. The authors addressed MYD88 dependent and independent pathways. However, the current study only detected the expression of the components of these pathways, which is not sufficient to indicate activity. The authors may do some discussions according to the current data, but putting these pathways in the conclusion is too stretchy.

We have edited our discussion to note that AS101 and MS275 act on the transcription of MyD88 Dependent and independent pathway components. Lines 453-471:

“We also demonstrate that IL-10 and class 1 HDACs both independently and synergistically regulate IL-12b transcription, and that IL-10 modulates the TLR -NF-κB signaling pathway by suppressing the transcription of both MyD88-dependent and MyD88-independent pathways. The latter finding provides an additional means of IL-10 modulating its own transcription and secretion (Figure 1A and 6A), in addition to regulating its own receptor (52), or post-transcriptionally (53-55), and corroborates the finding by He et al that TLR4-NF-κB signaling differentially regulates IL-10 and IL-12b transcription, through regulation of c-fos synthesis and differential binding of the NF-κB transcription factor to the two cytokine promoters. AS101 is known to disrupt IL-10 secretion by inhibiting integrin activity (64). The loss of integrin activity likely explains the discrepancy observed between the transcriptional and secreted cytokine analysis. In addition, we showed that IL-10 and class 1 HDAC inhibition differentially affected IκBα synthesis and proteolytic degradation, shifting the TLR-NF-κB signaling to anti-inflammatory territory. Both enhance IkKα and RelA expression, but only IL-10 inhibition increases p50 gene expression. Lastly, we show that IL-10 inhibited the transcription of both MyD88-dependent and MyD88-independent pathway components, while class 1 HDAC inhibition only appears to affect the transcription of MyD88-dependent components. Figure 8A-B summarizes the effect of IL-10 and class 1 HDAC inhibition on the modulation of IκBα phosphorylation (AS101) and ubiquitination/degradation (MS275) in the TLR4 signaling pathway, and the proposed interaction between IL-10, IL-12b and class 1 HDACs with LPS stimulation.”

4. The authors need to explain IL-10 mRNA expression is not increased by LPS in figure 1A but protein is increased in figure 6A.

It is likely that IL-10 synthesis is regulated post-transcriptionally through a negative feedback loop mechanism, and we are planning a new project to study this specific topic in the very near future. The following was added to lines 315- 320:

“Of note, IL-10 mRNA transcription was noted to remain unchanged with LPS stimulation (Figure 1A), while protein synthesis increased (Figure 6A. It appears that anti-inflammatory IL-10 transcription takes place constitutively with an inhibitory process limiting protein secretion, and stimulation with LPS enables removes this inhibitory step to allow for greater protein synthesis. The topic of post-transcriptional IL-10 regulation is to be examined in a separate project.”

Sincerely, 

The author team.

---

## [Decision Letter · Decision Letter 2]

23 Dec 2020

IL-10 AND CLASS 1 HISTONE DEACETYLASES ACT SYNERGISTICALLY AND INDEPENDENTLY ON THE SECRETION OF PROINFLAMMATORY MEDIATORS IN ALVEOLAR MACROPHAGES

PONE-D-20-22628R2

Dear Dr. Kasotakis,

We’re pleased to inform you that your manuscript has been judged scientifically suitable for publication and will be formally accepted for publication once it meets all outstanding technical requirements.

Kind regards,

Partha Mukhopadhyay, Ph.D.

Section Editor

PLOS ONE

Additional Editor Comments (optional):

Reviewers' comments:

Reviewer's Responses to Questions

**Comments to the Author**

1. If the authors have adequately addressed your comments raised in a previous round of review and you feel that this manuscript is now acceptable for publication, you may indicate that here to bypass the “Comments to the Author” section, enter your conflict of interest statement in the “Confidential to Editor” section, and submit your "Accept" recommendation.

Reviewer #1: All comments have been addressed

2. Is the manuscript technically sound, and do the data support the conclusions?

Reviewer #1: (No Response)

3. Has the statistical analysis been performed appropriately and rigorously? 

Reviewer #1: (No Response)

4. Have the authors made all data underlying the findings in their manuscript fully available?

Reviewer #1: (No Response)

5. Is the manuscript presented in an intelligible fashion and written in standard English?

Reviewer #1: (No Response)

6. Review Comments to the Author

Reviewer #1: (No Response)

7. PLOS authors have the option to publish the peer review history of their article (what does this mean?). If published, this will include your full peer review and any attached files.

Reviewer #1: No

---

## [Editor Report · Acceptance letter]

11 Jan 2021

PONE-D-20-22628R2 

IL-10 AND CLASS 1 HISTONE DEACETYLASES ACT SYNERGISTICALLY AND INDEPENDENTLY ON THE SECRETION OF PROINFLAMMATORY MEDIATORS IN ALVEOLAR MACROPHAGES 

Dear Dr. Kasotakis:

I'm pleased to inform you that your manuscript has been deemed suitable for publication in PLOS ONE. Congratulations! Your manuscript is now with our production department. 

Kind regards, 

on behalf of

Dr. Partha Mukhopadhyay 

Section Editor

PLOS ONE